# False Friends? On the Effect of Bureaucracy, Informality, Corruption and Conflict in Ukraine on Foreign and Domestic Acquisitions

**Viktoriya Gonchar [1], Oleksandr Kalinin [1], Olena Khadzhynova [2] and Killian J. McCarthy [3,*]**

[1] Marketing and Business Administration Department, Pryazovskyi State Technical University (PSTU), Mariupol, 87500 Donetsk, Ukraine; gonchar.mariupol@gmail.com (V.G.); kalininandkalinin@gmail.com (O.K.)

[2] Department of Economic Theory and Business Undertakings, Educational-Scientific Institute of Economics and Management, Pryazovskyi State Technical University (PSTU), Mariupol, 87500 Donetsk, Ukraine; azsudcom@gmail.com

[3] Faculty of Economics and Business, University of Groningen, 9700 AE Groningen, The Netherlands

[*] Correspondence: k.j.mccarthy@rug.nl

**Abstract:** Ukraine had had its ups and downs in recent years. It has, for example, dramatically improved its ease of doing business (*EOBB*), and it has made some progress reducing the relative size and influence of its shadow economy (*Shadow*). But, the Russian invasion of 2014 (*Conflict*) forced it to take a few developmental steps backwards. In this paper, we consider the effect of these factors, positive and negative, on the number of mergers and acquisitions, involving Ukrainian firms. We construct a sample of 4030 acquisitions in the period 1 January 2000–31 December 2020. Our results suggest that while the number of acquisitions by domestic firms increases in efficiency (+*EOBB*), transparency (−*Shadow*) and peace (−*Conflict*), the number of foreign acquisitions increases in bureaucracy (−*EOBB*), in informality (+*Shadow*), and unrest (+*Conflict*). From an academic perspective, our findings fit with some recent work, while providing new insights too. From a policy perspective, our findings that the number of foreign acquisitions is negatively affected by Ukraine's attempts to modernize and improve its economy and is positively affected by the ongoing conflict with Russia, makes us wonders what type of 'false friends' make such investments.

**Keywords:** mergers and acquisitions; Ukrainian mergers and acquisition; merger motives; merger trends; institutional forces; ease of doing business index; shadow economy; conflict; Ukraine

## 1. Introduction

In 2021 Ukraine celebrated its 30th anniversary as an independent country. It is a young country, and as such it has 'one foot in the past'—a long history of bureaucracy and state control—'one foot in the present'—a large, corrupt, informal economy—and it has 'its eyes fixed on the future'—on membership of the European Union (EU).

Ukraine has taken significant steps towards achieving this goal. It has, for example, developed more efficient regulations and stronger protections for private property rights. As a result, Ukraine has climbed the World Banks' ease of doing business (EOBB) ranking, from 152nd out of 192 countries survey, in 2011, to 64th in 2020.

Ukraine has also made significant efforts to reign in its shadow economy. However, here the process is less impressive. The shadow economy in Ukraine is a result of "insufficient quality of state institutions... imperfection of the tax system... and corruption" (Vinnychuk and Ziukov 2013, p. 141). Ukraine has reduced the share of the shadow economy from just under 50% of total activity in 2000 to just over 30% in 2021. The decline is welcome—because the size of the shadow economy positively affects crime (e.g., Dreher and Schneider 2010; Schneider 2011) and negatively affects growth (e.g., Hodge et al. 2011) and investment (e.g., Eilat and Zinnes 2002)—and 30% is the international average (Medina

and Schneider 2018). Still, at 30% the shadow economy in Ukraine is far larger than, for example, the United States (7%), the Netherlands (7.6%), or China (12%). In fact, Ukraine stubbornly remains the most corrupt country on the continent after Russia.

Unfortunately, Ukraine has also been forced to take a few steps backwards on its developmental path. In 2014, Russia invaded Ukraine, annexed Crimea, which was one of the wealthier regions of the country, and sponsored separatists in the East. As a result, Ukraine's GDP dropped 15% (Olekseyuk and Schürenberg-Frosch 2019). Perhaps more damagingly, Russia also supported a 'frozen conflict', in Ukraine, which lasted until Russia re-invaded in 2022.[1] As a result, Ukraine not only lost its territorial integrity, but it lost its image too as a safe country, on a clear path to membership of the European Union.

In this paper, we consider the effects of these institutional factors—a clear step forward, in terms of an improving EOBB ranking, a more modest step forward in the fight to fully reign in the shadow economy, and a clear step backwards thanks to the Russian invasion— on the number of mergers and acquisitions involving Ukrainian firms.

We consider these effects on the number of mergers and acquisitions for two reasons. Firstly, mergers and acquisitions are an international market (McCarthy et al. 2016; McCarthy and Aalbers 2016), and research shows that they are sensitive to institutional changes (e.g., McCarthy and Dolfsma 2015). As such, mergers and acquisitions provide policy makers with insights into the market's views of the changes that a country makes. Second, there has been significant research on the trends in mergers and acquisitions (e.g., Andrade et al. 2001) as well as their performance (e.g., King et al. 2021), but the 'why' of the deal remains unclear (e.g., Aalbers et al. 2021). As a country that is, has, and is still undergoing significant institutional changes, Ukraine provides an excellent setting to understand which factors, together or individually, explain the 'why' of the acquisition. Thus, taken together, our aim in this study is to understand: what explains the number of mergers and acquisitions involving Ukrainian targets by foreign and domestic acquirers?

To answer this question, we collect data on all mergers and acquisition involving Ukrainian firms, from the Refinitiv (formally Thomson) SDC Database. We count the total number of acquisitions involving Ukrainian targets, in the period 1 January 2000–31 December 2020, and we distinguish between the number of acquisitions by foreign and by domestic acquirers. We then consider how: (1) the improvements in Ukraine's ease of doing business ranking; (2) the reduction in the relative size of the shadow economy; and (3) the start of the Russian conflict, in 2014 to the end of our data, affects each of these series.

Our results suggest that there are important differences between foreign and domestic acquirers of Ukrainian firms. We find, for example, that the number of acquisitions of Ukrainian targets by domestic acquirers increases in efficiency (+*EOBB*) and transparency (−*Shadow*), and peace (−*Conflict*), as could be expected. We find, however, that the number of acquisitions of Ukrainian targets by foreign acquirers does precisely the opposite. Our results suggest that the number of foreign acquisitions increases in bureaucracy (−*EOBB*), in informality (+*Shadow*), and in conflict (+*Conflict*). From an academic perspective, these are interesting results, which add insights into existing literature (e.g., Lee 2018). From a policy perspective, however, the finding that the number of foreign acquisitions is negatively affected by Ukraine's attempts to open, modernize, and clean its economy, is concerning; one wonders what type of 'false friends' make such investments in Ukraine.

Our paper is organized as follows. In the next section we provide additional insights into Ukraine, and on the literature that our paper builds upon. In Section 3 we describe the data that we collected, the variables that we programmed, and the analysis that we conducted. In Section 4 we present and describe our results. In Section 5 we describe the contributions of our study, we discuss the implications, the limitations, and the future research possibilities that these imply. Finally, we draw conclusions in Section 6.



## 2. Background

### 2.1. Ukraine

Modern Ukraine was founded in 1991, after the collapse of the Soviet Union. Today, it is the second largest country in Europe, after Russia. It is home to 42 million people, 72% of which are ethnic Ukrainian and 17% ethnic Russian. Ukraine is ranked 57th in terms of GDP and it is classed, therefore, as a developing country. Its primary industries are low-value, high-volume industries, such as agriculture, and materials, such as iron.

Over the last 30 years since it was founded, Ukraine has made significant efforts to open and liberalize its economy. For example, it has increasingly turned its back on its bureaucratic, state-run past, to embrace a more open, efficient, and transparent business environment. This has led Ukraine to rise by almost 100 places on the World Bank's ease of doing business rankings, from 152nd out of 190 countries, in 2011, to 61st by 2020. It has tried to tackle corruption too, although here there is much work to be done; Ukraine remains the second most corrupt country on the European continent (Transparency International 2022). Ukraine has, however, made significant progress formalizing its economy. It has reduced the shadow economy from over 50% of total economic activity in 2000 to just over the international average of 30% in 2020 (Medina and Schneider 2018). At 30%, however, the shadow economy in Ukraine is significantly higher than developed countries, such as the United States (7%), the Netherlands (7.6%), or the United Kingdom (8.3%), it is much higher than Ukraine's neighbours, such as Poland (16.6%), and it is significantly higher than other developing countries, such as China (12%), too.

Ukraine's progress was dramatically interrupted in 2014 when Russia invaded Ukraine, annexed Crimea, and supported separatists in Donetsk and Luhansk. The separatists declared independence from Ukraine and, with Russian support, maintained a 'frozen conflict' with the Ukrainian army, in the latter period of our analysis (2014–2020). The Russians invaded again in 2022, ostensibly to support the separatists. The reality, however, is that the 2022 invasion—ongoing at the time of writing and thus beyond the scope of our paper—impacted all of Ukraine. As of April 2022, the invasion led 4 million refugees to leave the country, while another 6 million were internally displaced, and it caused material damages to the Ukrainian economy in the range of USD 543–600 billion.[2]

### 2.2. The Market for Ukrainian Firms

We consider the effects of the changes described above—that is, increasing ease of doing business, a shrinking shadow economy, and the outbreak of the conflict with Russia in 2014—on the number of mergers and acquisitions of Ukrainian target firms.

We consider the effects on the number of mergers and acquisitions of Ukrainian target firms for two reasons. First, and from the perspective of Ukraine, the market for merger and acquisitions is an international market (e.g., McCarthy and Aalbers 2016; McCarthy et al. 2016), and research shows that it is sensitive to institutional factors (e.g., McCarthy and Dolfsma 2015). As such, the number of mergers and acquisitions of Ukrainian firms can be used as a barometer, to measure the market's views of Ukraine, the changes it has undergone, and the challenges it faces, which will be of interest to policy-makers.

Second, and from the academic perspective, there has been significant research on general acquisition trends (e.g., Andrade et al. 2001) and on acquisition performance in particular (e.g., King et al. 2021). The 'why' of acquisitions, however, remains unclear (e.g., Aalbers et al. 2021). As such, Ukraine, the changes it has undergone, and the challenges it faces, and the effects of all of this on the number of mergers and acquisitions announced, provides academics with insights into which factors explain the 'why' of the deals.

Turning to the three institutional factors, we consider the effect changes in the ease of doing business (EOBB) on the number of mergers and acquisitions of Ukrainian firms. A higher EOBB ranking indicates better, usually simpler, regulations for businesses and stronger protections of property rights. Unsurprisingly, research shows that the level of foreign investment increase with a counties' ranking (e.g., Corcoran and Gillanders 2015; Doshi et al. 2019). Consequently, we predict that Ukraine's rise, from 152/190 to

64/190 in terms of EOBB ranking, will have a positive effect on the number of mergers and acquisitions of Ukrainian firms. Theoretically, and as all acquirers should benefit from a decrease in the level of bureaucracy in the country, we do not see any reason to suggest that the effect should differ for foreign or domestic acquirers.

Next, we consider the effect of Ukraine's efforts to reign in the shadow economy. The shadow economy in Ukraine is the result of "insufficient quality of state institutions, ineffective regulatory policy, imperfection of the tax system, unformed competitive environment and corruption" (Vinnychuk and Ziukov 2013, p. 141). The fact that share of the shadow economy plummeted from just under 50% in 2000 to just over 30% in 2021, as Ukraine formalized its economy, is welcome. Research on the shadow economy (e.g., Dreher and Schneider 2010; Schneider 2011) positively links the size of the shadow economy to corruption and criminal activities, like money laundering (e.g., McCarthy et al. 2015; Gnutzmann et al. 2010), and negatively links it to economic growth (e.g., Hodge et al. 2011), as well as foreign investment (e.g., Eilat and Zinnes 2002), and mergers and acquisitions (e.g., Cuong et al. 2021). Consequently, we predict that as the size of the shadow economy in Ukraine declines, the number of mergers and acquisitions in Ukraine will increase. Again, as everyone benefits from a formal economy, we do not see any theoretical reason to suggest that the effect should differ for foreign or domestic acquirers.

Finally, we consider the influence of the conflict with Russia, which annexed Crimea in 2014, sponsored separatists to maintain a 'frozen conflict' in Donbas between 2014 and 2022, before launching a full-scale invasion in February 2022. It goes without saying that a full-scale invasion in 2022 will be 'bad for business'; the IMF predicts that Ukraine's GDP in 2022 will drop by as much as 35%. In this paper, however, we focus on the effect of the 'frozen conflict' in the period 2014–2020. Research argues that an increase in risk will reduce investment (e.g., Kiymaz 2009), as managers become more cautious. However, research also suggests that risk can strengthen the hand of the acquiring firm (e.g., Lee 2018), which might lead to more 'bargain' deals, and actually increase the number of foreign acquisitions. Thus, it is unclear what the effects of the conflict might be.

## 3. Methods

### 3.1. Sample

We used Refinitiv (formerly Thomson) SDC to collect the data necessary to test our hypothesis. We refined it to include: (1) all mergers and acquisitions; (2) involving a Ukrainian firm; (3) in which 100% of the target firm was acquired; (3) in the period 1 January 2000–31 December 2020. Doing so we generate a sample of 4225 mergers and acquisitions.

### 3.2. Variables

We create a number of variables to test our hypotheses. In this section we describe the three dependent variables, three independent variables, and the controls that we use.

Dependent variable(s): We created three *Count* dependent variables for our analysis. Per year, we estimated: (1) the total number of acquisitions involving Ukrainian targets; (2) the number of acquisitions involving Ukrainian targets, by domestic acquirers; (3) the number of acquisitions involving Ukrainian targets, by foreign acquirers.

Independent variables: We created three independent variables for our analysis. First, we collected the ease of doing business (EOBB) data on Ukraine from the World Bank. That data is available for the period 2006–2020. We used this to programme an *EOBB* variable. We set this as the inverse of the *EOBB* ranking, to facilitate interpretation. Second, we collected data on the shadow economy in Ukraine, as reported by the Ukrainian Ministry for the Economy (see Figure 1). We used this to create a *Shadow* variable. We set this equal to the size of the shadow economy as a proportion of total economic activity. Finally, we created a *Conflict* dummy, to distinguish between the period before and after the 2014 Russian annexation of Crimea. We set the *Conflict* dummy equal to 0 in the period before the start of the conflict with Russia, and equal to 1 in the period thereafter.

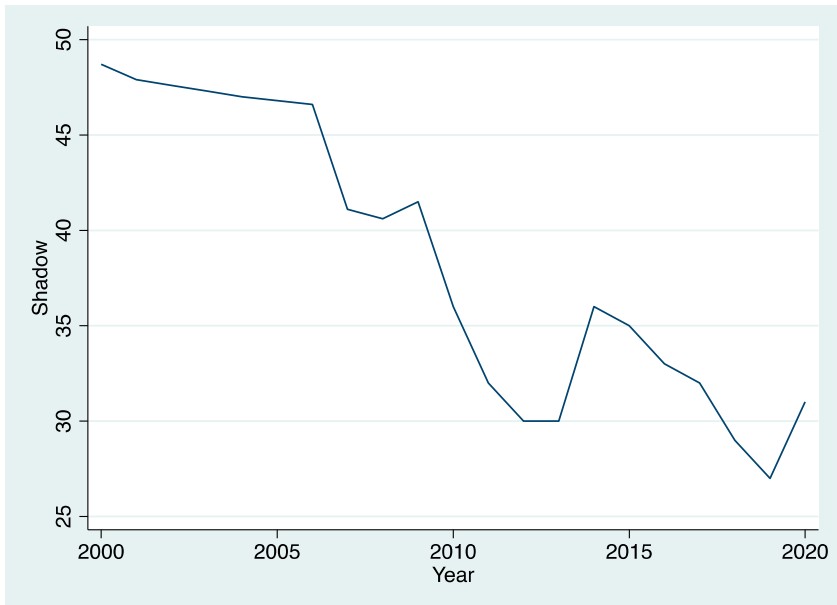

**Figure 1.** Share of the shadow economy in the total economy over time.

Control variables: We used Ukrainian GDP, as reported by the World Bank, to control for general economic activity, and year dummies to control for year-specific effects.

Table 1 provides an overview of the variables that we employed in our study. It names the variables, describes how they are constructed, and shows their values too.

**Table 1.** Overview of the Key Variables and their Definition.

| Variable | Definition | Value |
|---|---|---|
| *Count* | Number of acquisitions involving Ukrainian targets: (1) in total; (2) by domestic acquirers (3) by foreign acquirers. | Count |
| *EOBB* | Inverse of the national rank of ease of doing business (EOBB) | $-1$ to $-190$ |
| *Shadow* | Size of the shadow economy divided by the total economy | 0.00–1.00 |
| *Conflict* | Dummy set to 0 prior to 2014 and 1 thereafter | 1 or 0 |
| GDP | Ukrainian real GDP | USD billions |
| Year | Year dummy | 1 or 0 |

### 3.3. Estimation Strategy

Our dependent—the number of acquisitions—is a count variable. It only takes non-negative integer values and has a skewed distribution. For such a dependent, a linear regression model would result in inconsistent, biased, and inefficient estimates (Greene 2003; Hausman et al. 1984). A Poisson or negative binomial regression could be used. Since Poisson makes a strong assumption regarding equal mean and variance—which is not true in our case—we select a negative binomial model to test our hypotheses.

### 3.4. Regression Model

We estimate the following equation:

$$Count_t = \alpha + \beta_1 EOBB_t + \beta_2 Shadow_t + \beta_3 Conflict_t + \beta_4 GDP_t + \lambda_t + \varepsilon_t$$

where: (1) $Count_t$ is either (i) the total number of acquisitions involving Ukrainian targets, (ii) the number of acquisitions involving Ukrainian targets by domestic acquirers, or (iii) the number of acquisitions involving Ukrainian targets by foreign acquirers, in year $t$; (2) $EOBB_t$ is the inverse of the national ease of doing business score, in year $t$; (3) $Shadow_t$ is the share of the shadow economy, in year $t$; (4) $Conflict_t$ is a dummy, to distinguish between the period before and after the 2014 Russian annexation of Crimea; (5) $GDP_t$ is a control, to account for total economic activity, in year $t$; (6) $\lambda$ is a set of year dummies to control for year-specific effects; and (7) $\varepsilon_t$ is a normally distributed error term.

## 4. Results

### 4.1. Descriptives

There were 4225 mergers and acquisitions, in our sample of acquisitions involving Ukrainian firms, in the period 1 January 2000–31 December 2020. In 95% (4030) of cases, the Ukrainian firm was the target. Our sample includes 2268 acquisitions of Ukrainian firms by foreign acquirers and 1761 by domestic acquirers. We focus on these—4030 acquisitions in which the Ukrainian firm was the target—for the remainder of our analysis.

Figure 2 documents the total number of acquisitions of Ukrainian firms, per year, as well as the total by acquirer type. The rise in activity in the boom of 2005–2010, and the sharp decline of 2014, are clearly apparent. In total, the Ukrainian firms in our sample were acquired by acquirers from 66 different countries. Cyprus (540)—which is home to a large Ukrainian and Russian business community—and Russia (247)—which is, of course, historically economically and linguistically close to Ukraine—were the two biggest acquirers. These were followed by The Netherlands (108), the United States (101), and the United Kingdom (89). Figure 3 is a heatmap that illustrates the spread of the acquirers in our sample, by GPS location. It shows that, in terms of cities, the acquirers of Ukrainian firms are often concentrated in places such as Brussels, Zurich, Budapest, and Riga.

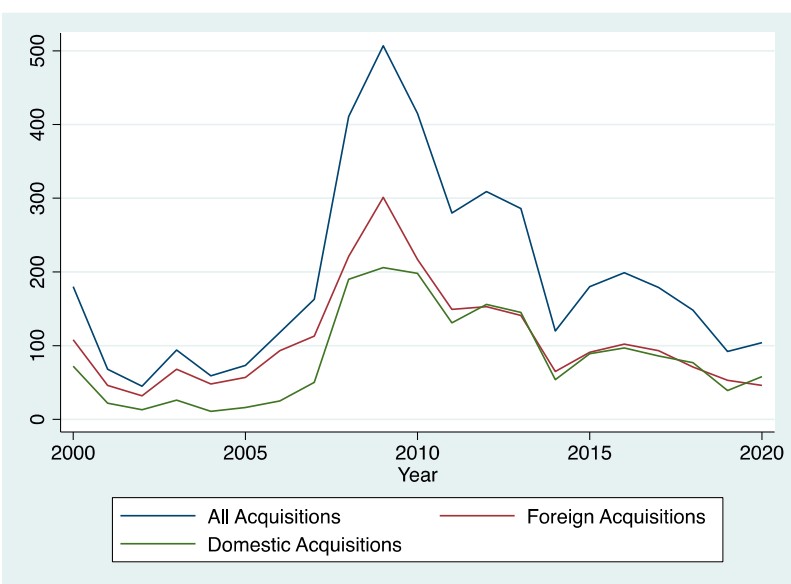

**Figure 2.** Number of acquisitions, per year, in total, and per acquirer type.

Table 2 provides a break-down of the Ukrainian targets by industry. It reports that 20% (808) of acquisitions involving Ukrainian targets involve firms in the financial industry, 19.5% (786) are from consumer staples, and 14% are from the industrial segments. The same pattern is apparent when looking at acquisitions by foreign or domestic acquirers. Thus, 'what' is being bought does not differ for foreign and domestic firms.

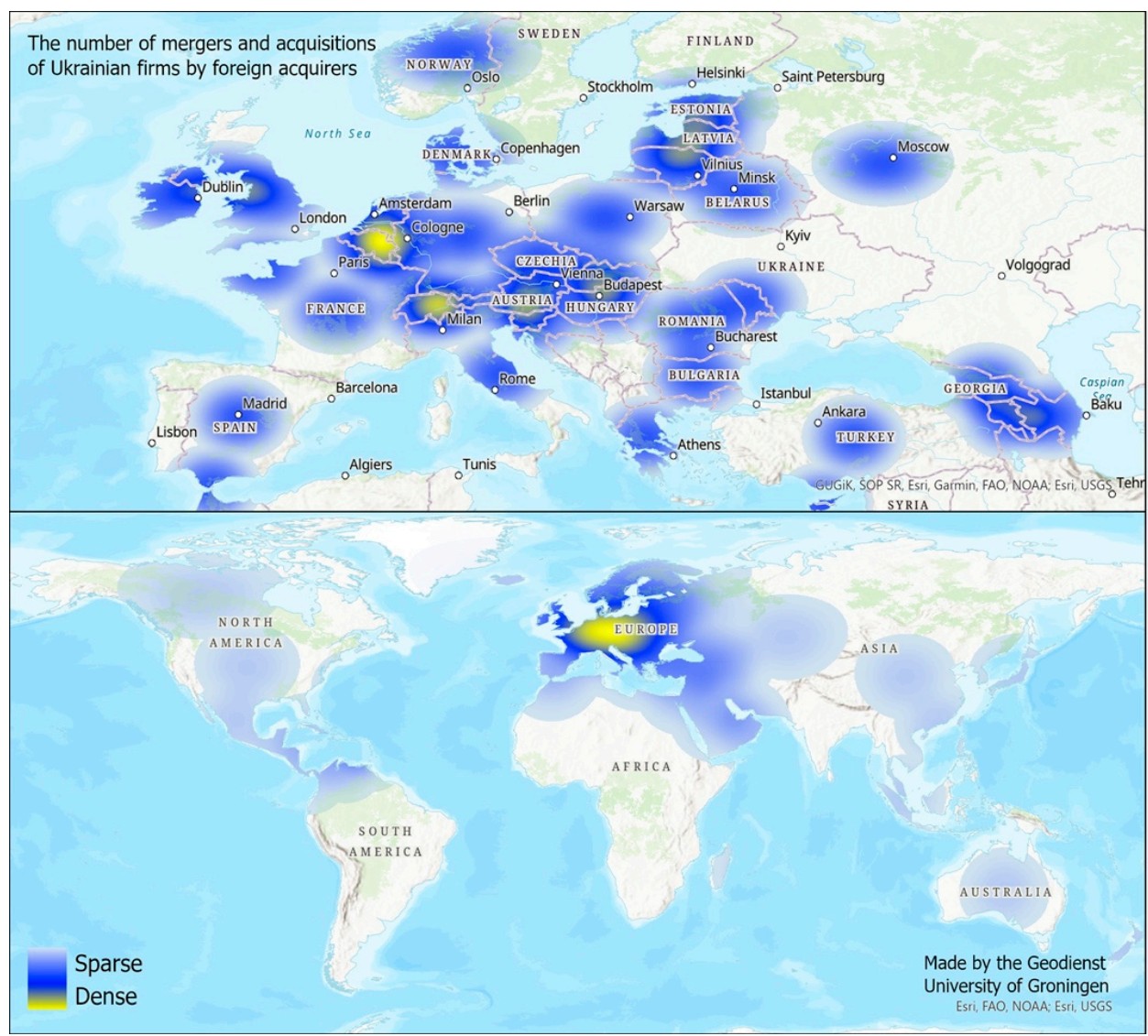

**Figure 3.** Distribution of foreign acquirers of Ukrainian firms.

**Table 2.** Ukrainian acquisitions by industry, in total, and per acquirer type.

| Industry Segment | Number of Deals Involving Ukrainian Targets (%) | | |
|---|---|---|---|
| | **All** | **Foreign Acquirers** | **Domestic Acquirers** |
| Consumer Products | 134 (3.3) | 77 (3.4) | 57 (3.2) |
| Consumer Staples | 786 (19.5) | 372 (16.4) | 414 (23.5) |
| Energy and Power | 519 (12.8) | 323 (14.2) | 196 (11.1) |
| Financials | 808 (20.8) | 411 (18.1) | 397 (22.5) |
| Government and Agencies | 5 (0.12) | 1 (0.04) | 4 (0.02) |
| Healthcare | 61 (1.5) | 42 (1.85) | 19 (1.07) |
| High Technology | 101 (2.5) | 64 (2.82) | 37 (2.10) |
| Industrials | 574 (14.2) | 346 (15.26) | 228 (12.94) |
| Materials | 548 (13.6) | 347 (15.30) | 201 (11.4) |
| Media and Entertainment | 151 (3.7) | 68 (3) | 83 (4.7) |
| Real Estate | 127 (3.1) | 83 (3.6) | 44 (2.4) |
| Retail | 117 (2.9) | 72 (3.1) | 45 (2.5) |
| Telecommunications | 99 (2.4) | 62 (2.7) | 37 (2.1) |
| Total | 4030 (100) | 2268 (100) | 1761 (100) |

Finally, Table 3 provides details on a number of measures which are useful to understand the size of the firms involved in the sample, and the size and type of the deals that they concluded. It reports, for example, that the average acquirer in the sample had 46,890 employees, and had an enterprise value of USD 145 million. The average deal in the sample was for a 62% share of the target firm, and the average deal was worth USD 76 million. These figures should be treated with caution, however, given the low number of observations. For example, there is only data for deal value in 572 (14%) of the deals in our sample.

**Table 3.** Acquirer and acquisition size.

| Variable | Obs | Mean | Std. Dev. | Min | Max |
|---|---|---|---|---|---|
| Deal Value | 575 | 76.60253 | 419.572 | 0.012 | 5515.84 |
| No. Employees | 329 | 46,890.35 | 115,883.4 | 2 | 622,000 |
| Enterprise Value | 447 | 145.041 | 589.9543 | −167.74 | 7881.409 |
| Percent Sought | 3339 | 62.09998 | 33.84988 | 0.001 | 100 |

### 4.2. Acquisition of Ukrainian Firms

Table 4 reports results for five negative binomial regression models. In each case, the dependent is the per year total number of acquisitions involving Ukrainian firms.

**Table 4.** Number of deals involving Ukrainian targets.

| | Model 1 | Model 2 | Model 3 | Model 4 | Model 5 |
|---|---|---|---|---|---|
| *EOBB* | | 0.026 ***<br>[0.000] | | | 0.116 ***<br>[0.000] |
| *Shadow* | | | 0.031 ***<br>[0.000] | | −0.148 ***<br>[0.000] |
| *Conflict* | | | | −9.772 ***<br>[0.000] | −6.577 ***<br>[0.000] |
| *Year Dummies* | Included | Included | Included | Included | Included |
| *GDP* | −0.004 ***<br>[0.000] | −0.035 ***<br>[0.000] | −0.000 ***<br>(0.000) | 0.074 ***<br>[0.000] | −0.059 ***<br>[0.000] |
| *Pseudo $R^2$* | 0.433 | 0.414 | 0.433 | 0.433 | 0.414 |
| *AIC* | 186.542 | 113.371 | 144.542 | 144.542 | 109.371 |
| *BIC* | 208.477 | 115.495 | 144.542 | 144.542 | 110.079 |
| *Ll* | −72.271 | −53.685 | −72.271 | −72.271 | −53.685 |

*** = $p < 0.01$.

Model 1 is a base model: it shows that there is a negative and significant effect between the number of deals and GDP, controlling for year specific effects. This would seem to imply that the number of mergers and acquisitions in Ukraine increases as the economy contracts, which fits well with the conclusion that mergers and acquisitions, announced in periods of economic expansion, destroy value (e.g., Moeller et al. 2005).

Model 2 adds *EOBB* to this base model specification to consider the effects of Ukraine's rise in the ease of doing business rankings. It shows that there is a positive and significant effect, which means that, as Ukraine becomes an easier place to do business, the number of deals that are done, involving Ukrainian firms, increases.

Model 3 replaces *EOBB* with *Shadow*, the shadow economy measure. It reports a positive and significant relationship between the size of the shadow economy and the number of deals announced. Put another way, the results of model 3 suggests that as Ukraine matures, and formalizes its economy and, in the process, reduces the relative

size of the shadow economy, the number of mergers and acquisitions of Ukrainian firms declines.

Model 4 considers the cost of the conflict with Russia on the number of deals. As expected, it reports a negative relationship between the start of the conflict in 2014, and the level of investment, in the form of mergers and acquisitions of Ukrainian firms.

Finally, model 5 presents a full model. It reports consistent results for both EOBB and for *Conflict*. In model 5, however, *Shadow* has a negative and significant effect on the number of mergers and acquisitions involving Ukrainian firms, when controlling for the effects of *EOBB* and *Conflict*. This means that, in contrast to the results presented in model 3, a decrease in the size of the shadow economy actually has a positive effect on the number of mergers and acquisitions of Ukrainian firms, once we account for *EOBB* and *Conflict*.

### 4.3. Acquisition of Ukrainian Firms by Domestic Acquirers

Table 5 reports results for another five negative binomial regression models. These models repeat the analysis described on Table 3, this time using the per year total number of acquisitions involving Ukrainian firms, by domestic acquirers.

**Table 5.** Number of deals involving Ukrainian targets by domestic acquirers.

|  | **Model 6** | **Model 7** | **Model 8** | **Model 9** | **Model 10** |
|---|---|---|---|---|---|
| *EOBB* |  | 0.075 *** [0.000] |  |  | 0.276 *** [0.000] |
| *Shadow* |  |  | 0.094 *** [0.000] |  | −0.334 *** [0.000] |
| *Conflict* |  |  |  | −30.074 *** [0.000] | −15.235 *** [0.000] |
| *GDP* | −0.002 *** [0.000] | −0.076 *** [0.000] | 0.012 *** [0.000] | 0.240 *** [0.000] | −0.118 *** [0.000] |
| *Year Dummies* | Included | Included | Included | Included | Included |
| *Pseudo $R^2$* | 0.448 | 0.415 | 0.448 | 0.448 | 0.415 |
| *AIC* | 134.516 | 95.072 | 138.516 | 124.516 | 127.072 |
| *BIC* | 139.738 | 95.072 | 145.827 | 124.516 | 138.401 |
| *Ll* | −62.258 | −47.536 | −62.258 | −62.258 | −47.536 |

*** = $p < 0.01$.

The results on Table 5 are consistent with those presented in Table 4, although the effects are more pronounced. For example, comparing model 5 in Table 4 with model 10 in Table 5, we see that the coefficient on EOBB is twice as large in the domestic setting ($β = 0.276$, $p < 0.01$) than it is in the full sample. The same is true of *Shadow* (domestic: $β = −0.334$, $p < 0.01$; All: $β = −0.148$, $p < 0.01$), while the effects of *Conflict* are almost three times as large in the domestic set ($β = −15.235$, $p < 0.01$) than in the full set ($β = −6.577$, $p < 0.01$). This implies that the domestic market for Ukrainian targets is much more sensitive to the institutional changes that we consider in this paper, compared to the full market.

### 4.4. Acquisition of Ukrainian Firms by Foreign Acquirers

Finally, Table 6 repeats the analysis described above, this time using the number of acquisitions of Ukrainian firms by foreign acquirers as the dependent. Here, however, we see some important differences in terms of explaining the number of acquisitions.

**Table 6.** Number of Deals involving Ukrainian Targets by Foreign Acquirers.

|  | Model 11 | Model 12 | Model 13 | Model 14 | Model 15 |
|---|---|---|---|---|---|
| *EOBB* |  | −0.024 ***<br>[0.000] |  |  | −0.034 ***<br>[0.000] |
| *Shadow* |  |  | −0.031 ***<br>[0.000] |  | 0.025 ***<br>[0.000] |
| *Conflict* |  |  |  | 9.803 ***<br>[0.000] | 1.868 ***<br>[0.000] |
| *GDP* | −0.007 ***<br>[0.000] | 0.015 ***<br>[0.000] | −0.011 ***<br>[0.000] | −0.086 ***<br>[0.000] | −0.003 ***<br>[0.000] |
| *Year Dummies* | Included | Included | Included | Included | Included |
| *Pseudo $R^2$* | 0.414 | 0.405 | 0.414 | 0.414 | 0.405 |
| *AIC* | 141.383 | 102.227 | 171.383 | 175.383 | 128.227 |
| *BIC* | 145.561 | 103.643 | 191.229 | 197.318 | 138.847 |
| Ll | −66.692 | −49.113 | −66.692 | −66.692 | −49.113 |

*** = $p < 0.01$.

Again, model 11 is a base model, which reports the effect the economy; it is in line with the findings of model 1 and model 5. Model 12 adds the *EOBB* variable to the base specification. Model 12 suggests that *EOBB* has a negative and significant effect on the number of acquisitions involving foreign acquirers. Model 13 replaces *EOBB* with *Shadow*. It reports a negative and significant relationship between the size of the shadow economy and the number of deals by foreign acquirers. Model 14 considers the effect of *Conflict* on the number of deals by foreign acquirers. It reports a positive relationship between Conflict and the number of foreign acquisitions. Finally, model 15 presents a full model. It confirms the results of the previous models but shows that, when controlling for *EOBB* and *Conflict*, *Shadow* has a positive and significant effect on the number of acquisitions.

Taken together, the results described in Tables 5 and 6 suggest that *EOBB*, *Shadow*, and *Conflict* have decidedly different effects on the number of acquisitions by domestic and foreign acquirers of Ukrainian targets. Table 5 suggests that the number of acquisitions by domestic firms grows as the economy becomes more efficient (+*EOBB*), transparent (−*Shadow*), and peaceful (−*Conflict*). However, Table 6 shows that the number of foreign acquisitions increase in bureaucracy (−*EOBB*), informality (+*Shadow*), and conflict (+*Conflict*).

## 5. Discussion

### 5.1. Discussion

In this paper we consider how an increasing ease of doing business, a large but shrinking shadow economy, and the outbreak of the conflict with Russia in 2014, affects the number of mergers and acquisitions of Ukrainian target firms, by foreign and domestic acquirers. Our results offer a number of important insights worth highlighting.

First, our results suggest that there is a negative relationship between GDP and the number of mergers and acquisitions of Ukrainian targets by domestic acquirers, international acquirers, as well as the combination of all acquirers. Curiously, this finding implies that the number of mergers and acquisitions in Ukraine increases as the economy contracts. As such, this fits well with the conclusion of academics that suggest that mergers and acquisitions, announced in periods of economic expansion, should be avoided, as they tend to destroy the most value (e.g., Moeller et al. 2005). Unfortunately for shareholders, Ukraine seems to be the 'exception' in this, where the 'rule' is typically to do more acquisitions, and to destroy more value, in the period of growth and expansion.

Second, and in line with expectations, we find that as the Ukrainian economic system has become more efficient and the economy has become more formal and transparent,

the total number of mergers and acquisitions, as well as the number of mergers and acquisitions by domestic firms, has grown. The positive relationship between EOBB on the number of mergers and acquisitions fits with existing work on EOBB and investment (e.g., Corcoran and Gillanders 2015; Doshi et al. 2019), while the positive relationship between a shrinking shadow economy and the number of mergers and acquisitions, fits with prior work linking the shadow economy to economic growth, in general, (e.g., Hodge et al. 2011), and to mergers and acquisitions, in particular (e.g., Cuong et al. 2021). We also find, unsurprisingly, that the Russian invasion has had a negative effect on the number of acquisitions by domestic firms too. Again, this fits with existing research which argues that an increase in risk will reduce both investment and performance (e.g., Kiymaz 2009). Taken together, we can conclude, that mergers and acquisitions of Ukrainian firms, in general, and mergers and acquisitions of Ukrainian firms by Ukrainian firms, in particular, are motivated by efficiency and transparency, and they require political stability.

Given our finding that both foreign and domestic acquirers target the same sorts of Ukrainian firms, active in financial, consumer staples, and industrial industries, it would not be surprising if foreign acquirers followed the same pattern as domestic acquirers. We find, however, that foreign acquirers of Ukrainian firms follow a very different logic. Our results suggest that the number of acquisitions by foreign acquirers is negatively affected by improvements in efficiency, measured in terms of EOBB, and negatively affected by reductions in the size of the shadow economy. The precise reasons for why remain unclear. Given the positive relationship between the size of the shadow economy and the level of corruption and criminal activities (e.g., Dreher and Schneider 2010; Schneider 2011), it is conceivable that foreign acquirers are attracted to Ukraine precisely for the lower levels of efficient regulation, taxation, transparency, and higher levels of corruption. Interestingly, our results also suggest that the conflict with Russia has had a positive effect on the number of acquisitions by foreign acquirers. This finding is in line with Lee (2018), who suggested that political instability can strengthen the acquirers bargaining power, and reduces costs, which, relatively speaking, could make Ukrainian investments more attractive. Conflict, therefore, may mean that foreign acquirers can do cheaper deals.

### 5.2. Contributions and Implications

Our research is relevant to academics, to policy-makers, and to managers, and it makes a number of contributions to each field, which we will highlight in this section.

First, our study will be of interest to academics, interested in understanding general acquisition trends (e.g., Andrade et al. 2001; Andriuškevičius and Štreimikienė 2021) and performance (e.g., King et al. 2021; Krishnan and Wu 2022). Our work also adds to the limited evidence on the mergers and acquisitions in Ukraine (Maksymenko 2018). Our main contribution to the literature on mergers and acquisitions, however, will be to those interested in understanding the 'why' of merger and acquisitions (e.g., Aalbers et al. 2021; Chatterjee 1986; Devos et al. 2009; Rabier 2017; Niemczyk et al. 2022). Our results show that foreign and domestic acquirers of Ukrainian firms react differently to the changes Ukraine has undergone, in the last 30 years, as well as to the challenges it faces today. This is an important finding that is, to the best of our knowledge, new to the literature.

Second, our study will be of interest to policy-makers, because it describes the market's views of Ukraine, the changes it has undergone, and the challenges it faces. Our findings offer Ukrainian policy-makers a number of important insights. For example, our results suggest that policy makers can increase the level of domestic investment by further improving the ease of doing business, and by tackling the scale of the shadow economy. Unsurprisingly, ending the conflict with Russia will also positively affect domestic investment and will rebalance the scales when it comes to the bargaining power of foreign acquirers. Perhaps more importantly, our results suggest that policy makers should pay more attention to foreign acquirers of Ukrainian firms. Our results suggest that the number of foreign acquisitions is negatively affected by Ukraine's attempts to modernize and improve its

economy, and positively related to ongoing conflict with Russia; one wonders what type of false friends make such investments, at such times, and for such reasons.

Finally, our study will be of interest to managers of Ukrainian firms, who may, one day, find themselves as someone else's acquisition target. Our findings suggest that these managers should be highly sceptical of the motives of foreign acquirers in particular, when evaluating the attractiveness of the proposals that they are presented with.

*5.3. Limitations and Future Research*

Like all research, our research is subject to a number of limitations. We explore them in this section, and discuss the potential avenues for future research that they imply.

First, our data is collected from the SDC. While one of the best databases for research on mergers and acquisitions, the SDC is an American database with an American focus. We hope that future researchers will replicate our study using other databases.

Second, we count the number of mergers and acquisitions involving Ukrainian firms, but only count 100% acquisitions as acquisition. We do this in order to set boundaries. However, there may be interesting dynamics in acquisitions for less than 100%, which we may have missed. We hope that future researchers will investigate that possibility.

Third, we treat Ukraine as a whole country, and assume that EOBB, the share of the shadow economy, and the impact of the conflict with Russia, in the period of our analysis, are equal across the country. Ukraine is, however, a hugely diverse country. It is conceivable, therefore, that things such as the size of the shadow economy will vary per region. It is certain that the effect of the 2014 conflict, which was entirely in the south and east of the country—will have the regions in which it was centred, to a far greater degree than it will have impacted places such as Lviv, 1300+ kilometres to the west. We hope that future researchers will identify such within-country differences and will investigate their effects on the number of mergers and acquisitions, by domestic and foreign firms.

Finally, we study the period 1 January 2000—31 December 2020, and we comment on the impact of the Russian invasion of 2014, on the number of mergers and acquisitions. As we noted elsewhere in this paper, the conflict was 'frozen' in the period 2014–2020—in other words, in the latter quarter of our dataset—but the conflict became 'unfrozen' when Russia invaded again in 2022. The effect of that invasion—which is ongoing at the time of writing—is clearly beyond the scope of our data. We hope that future researchers will consider the effect of that invasion and, in so doing, will distinguish the effects of the 'frozen' and 'unfrozen' periods of the conflict on the number of acquisitions announced. To do so would be interesting from an academic perspective, but also from a policy perspective, as it would provide insights into the (opportunity) cost of conflict on the market.

## 6. Conclusions

The aim of this paper was to explain the number of mergers and acquisitions, involving Ukrainian targets, by foreign and domestic acquirers. To do so is important, we argued, because it helps policy makers, on the one hand, to quantify the effects of the institutional changes that they preside over, while helping academics, on the other hand, to understand the 'why' of mergers and acquisitions, by providing an opportunity to study how a variety of institutional factors affect the number of deals announced.

Our results show that there are significant differences between domestic and foreign acquirers. We find, for example, that domestic acquirers are encouraged by efficiency in the market—which we measure in terms of Ukraine's ease of doing business ranking—and by the formalizing of the economy—which we measure using the share of the shadow economy. Unsurprisingly, we find that domestic acquirers have also been discouraged, by the 'frozen conflict' with Russia, in the east. We find that the number of acquisitions by foreign acquirers, however, is driven by a very different dynamic. We find that the number of foreign acquisitions is negatively affected by improvements in Ukraine's ease of doing business, negatively affected by the Ukraine's attempts to formalize an ever-greater share of its economy, and are positively affected by the conflict with Russia.

In doing so, our research adds to a number of academic discussions on, for example, the ease of doing business, the shadow economy, the role of conflict in management, as well as the general literatures on mergers and acquisitions, and the more specific literatures on acquisition motives and Ukrainian mergers and acquisitions. Our findings raise important policy questions. For example: what type of foreign firms prefer to make their acquisitions, in bureaucratic, opaque, corrupt, and unstable institutional environments? At the very least, this would appear to be the behaviour of some 'false friends'.

**Author Contributions:** All authors contributed equally and all authors were involved in the planning, data collection, analysis, write up, and review process. All authors have read and agreed to the published version of the manuscript.

**Funding:** This research received no external funding.

**Institutional Review Board Statement:** Not applicable.

**Informed Consent Statement:** Not applicable.

**Data Availability Statement:** The data is available upon request to the corresponding author.

**Conflicts of Interest:** The authors declare no conflict of interest, beyond the fact that 3 of the 4 authors are directly linked to Ukraine, both by birth and by academic affiliation.

## Notes

1   Ukraine and the West refers to the events of 2014 and 2022 as 'invasions' and 'wars', while Russia, refers to them as 'liberations' and 'special operations'. We use the term 'conflict' to avoid political discussions that are otherwise beyond the scope of our paper.

2   At the time of writing, the Kyiv School of Economics, to which one of the one of the authors is affiliated, and the Ukrainian Ministry of the Economy estimate that the 2022 invasion led to losses in the range of USD 543–600 billion (see Kyiv School of Economics 2022).

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
