# Peer review of "False Friends? On the Effect of Bureaucracy, Informality, Corruption and Conflict in Ukraine on Foreign and Domestic Acquisitions"

_jrfm, doi:10.3390/jrfm15040179_

Round 1
Reviewer 1 Report
The article addresses a very interesting topic, the ideas are presented concisely, in a logical order. The manuscript has the potential to be published, but needs to be substantially improved in order to be published.
The main recommendations are
- Insert a Literature review section in which to present similar studies such as….a) Krishnan, C. N. V., & Wu, J. (2022). Market Misreaction? Evidence from Cross-Border Acquisitions. Journal of Risk and Financial Management, 15(2), 93. b) Niemczyk, J., Sus, A., Borowski, K., Jasiński, B., & Jasińska, K. (2022). The Dominant Motives of Mergers and Acquisitions in the Energy Sector in Western Europe from the Perspective of Green Economy. Energies, 15(3), 1065. c) Ahluwalia, S., & Kassicieh, S. (2021). Effect of Financial Clusters on Startup Mergers and Acquisitions. International Journal of Financial Studies, 10(1), d) Andriuškevičius, K., & Štreimikienė, D. (2021). Developments and trends of mergers and acquisitions in the energy industry. Energies, 14(8), 2158. e) González-Torres, T., Rodríguez-Sánchez, J. L., Pelechano-Barahona, E., & García-Muiña, F. E. (2020). A systematic review of research on sustainability in mergers and acquisitions. Sustainability, 12(2), 513.
- Authors may insert in this literature review section the results of other studies published in MDPI journals or other journals.
- The authors should comment in the discussion section on the results obtained in the context of other studies that confirm or refute the conclusions obtained.
- I recommend the extension of the conclusions section considering the quality of the scientific approach.
- In the conclusions section, the authors must present the limits of the research and the future directions of the research.
Author Response
Please see the uploaded word document

Reviewer 2 Report
The authors should consider the following recommendations in order to improve the original manuscript:
- The abstract should be more consistent with the main text of the paper, preferably structured, simple, specific, clear and unbiased.
- To include certain relevant research questions.
- The authors also did not provide sufficient evidence on literature review to support the hypotheses. The Introduction section also includes the Literature review section which is practically non-existent being mentioned only a few bibliographic references quite uncorrelated. Authors should take into consideration more current publications in the sphere of discussed subject matter, especially studies conducted during the last 5 years.
- On 24 February 2022, Russia started a military invasion in Ukraine. Indeed, it is an aggression, but not a war in the true sense. Moreover, 8 years ago, in February 2014, Russia also invaded Ukraine and has annexed the Crimean Peninsula. Authors research is focused on a time period before February 2022 so there is no connection to Russia's current military aggression in Ukraine. Moreover, the inclusion of the word "war" in the title seems irrelevant and outside the scope of the research, although it is a “hot” topic the authors approach is very unconvincing.
- There are serious errors and inadvertence regarding sample periods in the study of authors. For instance:
In the Introduction section “Specifically, we 45 collect data on the number of mergers and acquisitions, in which the target was a Ukrain firms, in the period 2000-2021.”
In the Results section, authors suggested that “There were 4,225 mergers and acquisitions, in our sample of acquisitions involving Ukrainian firms, in the period 2000-2020.”
- To develop the discussion of results and the conclusions resulting from conducted research.
- Deepen the description of the limitations of conducted research and indicate the trends for further empirical research.
- This research paper has almost 8 pages including first page and Reference section. All sections are either underdeveloped or completely missing. There are only 8 references.
- Authors do not follow JRFM standards for text citation and bibliographic references as recommended in Instruction for authors paper template.
- To expand the managerial implications in the article.
- I would also like to see a well-developed discussion comparing and contrasting solution/results presented in the work with existing work and then a subsection of it presenting contributions to theory/knowledge/literature and followed by a subsection on Implications for practice.
- Human proofreading, grammar and spelling correction are also required in order to improve the quality of the manuscript.
This article does not meet the publication standards in JRFM journal.
Author Response
Please see the word document

Reviewer 3 Report
This paper uses Ukraine as an example to discuss the “what” and ”why” of mergers and acquisitions. I think that the topic of this paper is interesting and meaningful. However, unless a major reversion is made, the current version of this paper will not be published.
- In the Abstract, the kind of method and data the authors used should be mentioned clearly. The reason is that this can help readers quickly master the main contents of this paper.
- In the Introduction, the background description of this paper is too weak.
- From line 29 to line 38, the authors only tell the readers the change of data, but without any data sources.
- The authors do not use any literature in the Introduction.
- Structure of this paper is missing.
- There is not literature review in this paper. How do the authors show the differences when compared with previous studies?
- The authors use an unbalanced data-set, does it match the models used in this paper.
- There is not model specification in this paper.
- Why the author only control GDP, how about firms and year, have the authors controlled them?
- Before Table 2, variable description including definition, unit and so on is needed. Otherwise, it’s hard for readers to understand the rest sections.
- From Table 3 to Table 5, the author only perform a description to empirical results. But the authors do not give any expansion, explanation or literature comparison of these results
- The Conclusion is also very weak. Only describing the findings is not enough. Implications, contributions, limitations, future directions should be concluded.
Good luck.
Author Response
Please see the word document, attached.

Round 2
Reviewer 1 Report
The authors extended and improved the manuscript according with recommendations of reviewers. I recommend again the extention of the references list.
Reviewer 2 Report
The original manuscript has been improved. The authors followed the recommendations included in the previous review report so that the quality of their research article has increased. I also appreciate the effort of the authors in this regards.
Reviewer 3 Report
The proposed revisions have been carried out.